# miR-3065-5p and miR-26a-5p as Clinical Biomarkers in Colorectal Cancer: A Translational Study

**DOI:** 10.3390/cancers16213649

**Published:** 2024-10-29

**Authors:** Berenice Carbajal-López, Antonio Daniel Martínez-Gutierrez, Eduardo O. Madrigal-Santillán, Germán Calderillo-Ruiz, José Antonio Morales-González, Jossimar Coronel-Hernández, Joey Lockhart, Oliver Millan-Catalan, Monica G. Mendoza-Rodriguez, Leonardo S. Lino-Silva, Germán Calderillo-Trejo, Ronen Sumagin, Carlos Pérez-Plasencia, Eloy Andrés Pérez-Yépez

**Affiliations:** 1Programa de Doctorado en Investigación en Medicina, Escuela Superior de Medicina, Instituto Politécnico Nacional, Ciudad de México 11340, Mexico; eb.carbajalopez@gmail.com; 2Laboratorio de Genómica, Instituto Nacional de Cancerología, Tlalpan 14080, Mexico; insomnium098@gmail.com (A.D.M.-G.); jossithunders@gmail.com (J.C.-H.); oliver.millan.sg@gmail.com (O.M.-C.); 3Laboratorio de Medicina de la Conservación, Escuela Superior de Medicina, Instituto Politécnico Nacional, Ciudad de México 11340, Mexico; eomsmx@yahoo.com.mx (E.O.M.-S.); jmorales101@yahoo.com.mx (J.A.M.-G.); 4Unidad Funcional de Gastroenterología, Oncología Médica, Instituto Nacional de Cancerología, Tlalpan 14080, Mexico; gcalderillo06@gmail.com (G.C.-R.); gcalderillo95@hotmail.com (G.C.-T.); 5Department of Pathology, Northwestern University Feinberg School of Medicine, 300 East Superior St., Chicago, IL 60611, USA; joey.lockhart@northwestern.edu (J.L.); ronen.sumagin@northwestern.edu (R.S.); 6Unidad de Biomedicina, Facultad de Estudios Superiores Iztacala, Universidad Nacional Autónoma de México, Tlalnepantla 04510, Mexico; monica.mendoza@iztacala.unam.mx; 7Departamento de Anatomía Patológica, Instituto Nacional de Cancerología (INCan), Ciudad de Mexico 14080, Mexico; saul.lino.sil@gmail.com; 8Laboratorio de Genómica, FES-Iztacala, Universidad Nacional Autónoma de México (UNAM), Iztacala, Tlalnepantla 54090, Mexico

**Keywords:** micro-RNA panel, colorectal cancer, prognosis, overall survival, translational study

## Abstract

Molecular biomarkers have gained a significant role not only in determining the prognosis but also the diagnosis of CRC in high-risk populations. Translational studies that evaluate the utility of molecular markers are fundamental to timely detection and prognosis stratification for treatment decisions in CRC. This study included a cohort of forty-nine CRC patients from the National Cancer Institute, Mexico, to evaluate the clinical utility of miRNA expression levels in CRC. We found that miR-3065-5p and miR-26a-5p levels discerned between CRC and non-tumoral tissues, and that high levels of miR-3065-5p were associated with better outcomes in CRC patients. Overall, miR-3065-5p expression levels and clinical stage remained as predictor factors of OS. Finally, the algorithm PROMIR-C (which included clinical and miR-3065-5p expression levels) was developed as a website to improve treatment decision-making and prognosis assessment. Altogether, these findings support the clinical utility of miR-26a-5p and miR-3065-5p in the diagnosis and prognosis of CRC.

## 1. Introduction

Colorectal cancer (CRC) is the third most frequent type of cancer and the second cause of mortality worldwide [1]. Several prognostic factors of overall survival (OS) have been described for CRC, including age, gender, laterality, microsatellite instability, and histological subsite. Nevertheless, the main prognostic factor of the disease is the clinical stage, and this is because the 5-year OS rate for early detection (clinical stages I–II) is up to 90%, while this rate drops drastically to 12.5% for late detection (clinical stage IV) [2,3]. Therefore, the establishment of new diagnostic and prognostic markers for timely detection of CRC represents a clinical priority, since this allows access to better therapeutic decisions, and therefore, an increase in overall survival rates [4,5]. Recently, micro-RNAs (miRNAs) have been established as potential tumor suppressors or promoters due to their essential role in the regulation of proliferation, metastasis, angiogenesis, autophagy, and apoptosis [6,7,8]. miRNAs are small non-coding RNAs composed of ~22 nucleotides that regulate messenger RNAs (mRNAs), inducing their degradation and/or preventing translation [9]. The potential of miRNAs as diagnostic and prognostic biomarkers has been demonstrated in various cancer types, including lung [10], breast [11], stomach [12,13], myeloma [14], acute myeloid leukemia [15], and testicular cancer, among others [16]. In CRC, miRNAs have been associated with carcinogenesis and progression, through the regulation of signaling pathways such as Hippo, Notch, and Wnt/β-catenin [17,18,19]. In addition, these molecules regulate inflammatory cytokine levels by transcriptional regulation of transcription factors such as NF-kB and STAT3 [20]. Furthermore, miRNAs have been proposed to be used as potential prognostic factors and therapeutic outcome predictors [21,22,23].

In previous work, we have established the putative role of miR-3065-5p, miR-335-5p, and miR-26a-5p in diagnosis and prognosis in CRC pre-clinical studies [24,25]. Using bioinformatic analyses, we identified a miRNA group with differential expression levels in CRC compared with healthy tissues. Specifically, the levels of miR-3065-5p and miR-335-5p were related to poor overall survival in CRC patients [25]. Furthermore, expression levels of miR-335-5p were higher in CRC compared to inflamed colon tissue in murine colitis vs. AOM/DSS CRC models. In addition, miR-26a-5p was described as an oncomir in CRC, due to its significant role in the progression of the disease and tumoral invasion [24]. Despite all of this evidence indicating miR-3065-5p, miR-335-5p, and miR-26a-5p as possible biomarkers for CRC [24,25], currently, no miRNAs are currently used in clinical practice due to the lack of validation in translational studies.

## 2. Materials and Methods

The present study is a retrospective, observational study of prospectively collected tissue samples and clinical data information of CRC patients, with Ethical Committee approval (approval number INCAN/CI/826/17) at INCAN.

### 2.1. Aim and Setting

In this study, we aim to evaluate the clinical utility of miR-3065-5p, miR-335-5p, and miR-26a-5p as molecular biomarkers in the diagnosis and prognosis of OS in a translational study involving a Mexican cohort of CRC patients treated at INCAN.

### 2.2. Clinical Tissue Samples and Data Collection

The analysis included forty-nine patients. Inclusion criteria consisted of men and women with a diagnosis of colorectal cancer, treated at INCAN between January 2016 and December 2022. Exclusion criteria included no tissue availability, incomplete clinical records, lack of at least 3 months of follow-up, and/or abandonment.

From the forty-nine patients (paraffin-embedded tissues), only thirteen non-tumoral adjacent tissues were available for analysis at the pathology department. Clinical data were collected from the patient’s registry at the Gastrointestinal Department of INCAN. Clinical variables included age, gender, tumor size, clinical stage (defined by the Eighth American Joint Committee on Cancer TNM system) [26], histological differentiation, laterality, metastasis site, surgical procedure, chemotherapy, radiotherapy, and follow-up status. The patients included in this study accepted and signed the informed consent form according to protocol approved by the Ethics Research Committee of the Institute (Approval number: INCAN/CI/826/17).

### 2.3. RNA Purification and miRNAs Expression Analysis

Total RNA was isolated using an RNA Easy FFPE extraction kit (cat. 73504 Qiagen, Hilden, Germany) following the manufacturer’s protocol. Then, cDNA of miR-335-5p, miR-3065-5p, and miR-26a-5p was synthesized using 100 ng of total RNA and a TaqMan Micro-RNA Reverse Transcription Kit (Applied Biosystems, Waltham, MA, USA). miRNA expression was detected by quantitative real-time RT-PCR (RT-qPCR) using an miRNA TaqMan probe (ID: MC10063, ID: MH18328, ID: 000405) and the Roche Light Cycler 2.0 (Roche, Basel, Switzerland). Amplification conditions were as follows: 10 min at 95 °C, followed by 40 cycles of 95 °C for 15 s, and 68 °C for 60 s. Relative expression of miRNAs was calculated by 2^−ΔCT^ analysis using RNU6 levels as an endogenous expression control.

### 2.4. Statistical Analysis

For the miRNA expression analysis cited in Section 2.3, values are expressed as the mean ± SEM. Data were analyzed using Prism 5.0 (GraphPad, Boston, MA, USA) software. For patients, statistical analysis was performed using SPSS v. 24 software. Continuous variables were expressed as mean ± standard deviation values or as median and range (minimum and maximum) values. Categorical variables were expressed as percentages. Statistical comparisons among groups were performed using the *t*-test when data were normally distributed; otherwise, the Mann–Whitney U test was applied.

### 2.5. Survival CRC Patients’ Analysis

Disease-free survival (DFS) was defined as the time from surgical procedure to recurrence of the disease, whereas overall survival (OS) was defined as the time from diagnosis of CRC to death from any cause. DFS and OS rates were calculated using the Kaplan–Meier method, and different variables were compared using the log-rank test. Cox regression was performed for univariate and multivariate analysis to identify variables that were predictors of OS. Statistically significance differences were assessed when *p* < 0.05 was bilateral.

### 2.6. Algorithm for Prognosis

The developed algorithm integrates clinicopathological features and miRNA expression levels to predict prognosis in CRC patients. By analyzing a cohort of forty-nine Mexican CRC patients, higher expression levels of miR-3065-5p correlated with poor OS, indicating its utility as a prognostic biomarker. The resulting algorithm, Prognosis miRNAs assessment in Cancer (PROMIR-C), incorporates miR-3065-5p expression levels along with clinical factors such as age, gender, tumor laterality, grade of differentiation, and clinical stage, resulting in a prediction score for prognosis in CRC.

## 3. Results

### 3.1. Flowchart for Tissue Samples

A total of one hundred and nineteen patients treated at the Gastrointestinal Functional Unit were selected for collection of paraffin-embedded tumor/non-tumoral adjacent tissue and RNA extraction from paraffin-embedded tissue. Seventy patients were eliminated because the tissue did not meet quality criteria and/or did not have enough RNA concentration. A total of forty-nine CRC tissues and thirteen non-tumoral adjacent tissues were selected for miRNA expression analysis by qPCR, clinicopathological analysis, DFS, and OS analysis considering miRNA expression levels (Figure 1).

### 3.2. Clinical-Pathologic Features of CRC Patients

A cohort of forty-nine patients with a diagnosis of CRC was included in the analysis to evaluate miR-335-5p, miR-3065-5p, and miR-26a-5p expression. Of the total, 55% (*n* = 27) were male, and the remaining 45% (*n* = 22) were female with a median age of 54.5 years (19–81). Regarding clinical stage, 65% (*n* = 32) were clinical stage III, and 35% (*n* = 17) were clinical stage IV. In terms of laterality, tumor location was described as on the left side in 61% (*n* = 30) of patients, and on the right side in 39% (*n* = 19). Regarding histological grade, 53% (*n* = 26) were poorly differentiated, 39% (*n* = 19) moderately, and 8% (*n* = 4) were well differentiated, and the most prevalent histological subtype was intestinal at 84% (*n* = 41). Regarding treatment, 94% (*n* = 46) had surgery and 95% (*n* = 47) received chemotherapy (Table 1).

### 3.3. miR-3065-5p and miR-26a-5p Expression Levels Discern Between CRC and Non-Tumoral Adjacent Tissues

To determine miRNAs with potential diagnosis utility, expression levels of miR-3065-5p, miR-26a-5p, and miR-335-5p were evaluated in CRC and non-tumoral tissue patients. miR-3065-5p levels were significantly lower in CRC as compared to non-tumoral adjacent tissues (median −3.62 vs. −7.78, *p* = 0.0362) (Figure 2A). In contrast, miR-26a-5p showed higher expression in CRC tissues compared to non-tumoral adjacent tissues (median 3.49 vs. 1.3, *p* = 0.0028) (Figure 2B). No differences in the expression levels of miR-335-5p were observed (*p* = 0.6230) (Figure 2C). These data validate the clinical utility of miR-3065-5p and miR-26a-5p in differentiating between CRC and non-tumoral tissue samples and highlights the potential implementation of miR-3065-5p and miR-26a-5p as clinical biomarkers for CRC diagnosis.

### 3.4. The Overall Survival of CRC Mexican Patients Is Associated with miRNA Expression Levels

To explore the potential predictive value of miR-3065-5p, miR-26a-5p, and miR-335-5p in OS in CRC, only CRC patients (*n* = 49) were considered. The median expression of miR-335-5p, miR-3065-5p, and miR-26a-5p was calculated and subdivided into two groups: (1) high and (2) low expression. Overall survival analyses were performed considering the expression levels of each miRNA. Patients with low expression of miR-3065-5p had a median OS of 70 months, while patients with high expression of the miRNA did not reach the median OS. A 5-year OS of 66% vs. 80% for low and high expression was observed (*p* = 0.041) (Figure 3A).

Moreover, a clinical tendency was observed in differences in OS analysis for miR-26a-5p and miR-335-5p. For both miRNAs, the low expression levels were associated with a median OS of 75 months, while patients with high expression levels did not reach the median OS. The 5-year OS was 66% vs. 75% for miR-26a-5p (*p* = 0.165) and 70% vs. 72% for miR-335-5p (*p* = 0.194) in relation to low and high expression levels, respectively (Figure 3B,C). Considering that the expression level of miR-3065-5p was a predictor of OS, a sub-analysis comparing the levels of this miRNA and clinical features such as gender, laterality, and clinical stage was performed (Figure 4A–C). Male patients with high expression of miR-3065-5p did not reach the median OS, versus 70 months for those with low expression (74% vs. 77%; *p* = 0.050). Nevertheless, the other clinical features did not show statistical differences. We also analyzed the association of the expression levels of these three miRNAs with DFS, but no differences were observed between the groups for miR-3065-5p, miR-26a-5p, and miR-335-5p (*p* = 0.689; *p* = 0.673; *p* = 0.879) (Appendix A). In addition, we conducted a uni-multivariate analysis including clinical features previously associated with the prognosis of CRC such as age, gender, clinical stage, grade of differentiation, and laterality. The results showed that clinical stage (HR: 1.30, CI 1.23–2.30; *p*: 0.001) and low expression levels of miR-3065-5p (HR: 1.30, CI 1.23–2.30; *p*: 0.001) remained as predictor factors of OS for CRC patients. These findings support the idea that the expression levels of miR-3065-5p have clinical utility in the prognosis of OS of CRC patients.

### 3.5. PROMIR-C (Prognosis miRNAs Assessment in Cancer) Clinical–Molecular Prediction Score for Prognosis in CRC Patients

The survival rate calculation algorithm, PROMIR-C (https://promir-c.tiiny.site/ (accessed on 20 October 2024)), systematically analyzes patient data and determines the percentage of individuals who survive based on demographic and clinical factors. The algorithm organizes the data into age groups, sex, tumor location, histological grade, histology, and clinical stage. Each category contains information on the number of patients who survived and those who did not. Subsequently, the algorithm iterates through each category, computing the survival rate by dividing the number of survivors by the total number of patients in that category. This process is repeated for all categories, resulting in survival rate percentages for each demographic and clinical subgroup. Additionally, the algorithm incorporates weighted factors to adjust for the relative significance of different characteristics in predicting survival outcomes. By iteratively analyzing and aggregating data from various categories, the algorithm provides a comprehensive overview of survival rates across different patient profiles, offering valuable insights for clinical decision-making and prognosis assessment.

In addition to the existing demographic and clinical variables, the algorithm now incorporates the miR-3065-5p variable to refine the calculation of survival rates. This additional variable, representing a specific genetic marker or biomarker, contributes valuable information regarding tumor biology and patient prognosis. The algorithm adjusts the survival rate computation to account for the expression levels (high or low) of miR-3065-5p, enhancing the accuracy and granularity of survival predictions. By considering this additional molecular factor alongside demographic and clinical characteristics, the algorithm provides a more comprehensive assessment of patient outcomes. Consequently, the algorithm generates two distinct survival rate percentages: one based on the original set of variables and another reflecting the inclusion of miR-3065-5p. This dual approach offers clinicians and researchers a nuanced understanding of survival probabilities, enabling more informed treatment decisions and prognostic evaluations in colorectal cancer care.

## 4. Discussion

miRNAs regulate a multitude of biological processes and play critical roles in carcinogenesis and progression of CRC [27,28,29,30]. Several studies have demonstrated the fundamental role of miRNAs in the diagnosis of CRC, due to their differential expression in healthy and CRC tissues [6,7,21,31]. Previously, using a bioinformatics approach, we demonstrated significant differences in the expression levels of miR-3065-5p, miR-26a-5p, miR-335-5p, miR-21-5p, miR-3405p, miR-577, miR-21-3p, miR-27b-5p, miR-335-3p, miR-215-5p, 30b-5p, miR-192-5p, and miR-432-5p between healthy and tumoral tissues [25]. Moreover, the expression of miRNAs in non-tumoral colon tissue has also been explored. Using mice models, we predicted that only the expression levels of miR-3065-5p, miR-26a-5p, and miR-335-5p could allow discrimination between healthy, inflamed, and CRC tissue [24,25]. Moreover, studies demonstrating the clinical utility of these miRNAs in the diagnosis or prognosis of CRC have not been previously developed. In this study, we identified differences in the expression levels of two miRNAs. A significantly low expression of miR-3065-5p and a higher expression of miR-26a-5p are distinctive in CRC tissues compared to non-tumoral tissues. These results are consistent with previous observations by Nadezhda et al., who demonstrated the anti-oncomir role of miR-3065-5p in melanoma. The downregulation of this miRNA was linked to suppressor effects on cell viability, proliferation, migration, and invasion capacity [32]. Additionally, miR-26a-5p has been demonstrated as a potential oncomir in papillary thyroid cancer [33] and CRC. In colon cancer, miR-26a-5p induces `the downregulation of PTEN gene expression levels, which triggers increased proliferation rates and enhances cell migration [24]. Due to the potential role of these miRNAs in carcinogenesis, the differences in the expression levels of miR-3065-5p and miR-26a-5p found in this work support their clinical utility in CRC diagnosis. The establishment of miR-3065-5p and miR-26a-5p as clinical biomarkers in the diagnosis of CRC will allow clinicians to distinguish these biomarkers in at-risk populations and improve early diagnosis and treatment.

Once a patient is diagnosed with CRC, the main prognostic factor is the clinical stage [34,35]. Patients diagnosed in early stages are treated with surgery alone, while patients diagnosed with advanced or metastatic disease require adjuvant/palliative chemotherapy or target therapy, which precludes quality of life and significantly reduces OS rates [3,36]. Besides the clinical stage, other clinicopathological features have been associated with the prognosis of CRC patients; these include gender, age, tumor location (laterality), grade of differentiation, histology, and microsatellite stability [5]. Nevertheless, CRC is a heterogenic disease with significant differences in prognosis and treatment response [37]. Thus, the addition of molecular tools that improve prognosis assessment and treatment decisions is essential [38,39]. miRNA expression levels have been linked not only with carcinogenesis but also with CRC prognosis [22,40,41,42,43]. Studies have demonstrated that miRNAs modulate the expression of genes to induce high rates of proliferation, angiogenesis, and metastasis, all biological functions essential in cancer progression [44]. Furthermore, several miRNAs were previously postulated as diagnostic and prognostic clinical biomarkers in colon cancer to refine clinical decisions in silico and cell lines [29]. Importantly, the clinical utility of miRNA expression levels in the prognosis of CRC-correlated clinical features remains unclear [45]. Therefore, translational studies remain fundamental to validate their clinical role in the diagnosis and prognosis of CRC. The evidence reported in this study establishes the crucial role of miR-3065-5p as a biomarker, not only for the diagnosis but also for the prognosis of CRC. In this prospective study that included a cohort of forty-nine patients diagnosed with CRC, we validated the clinical utility of miR-3065-5p as a prognostic marker of the CRC outcome. We found that high expression of miR-3065-5p is associated with better OS compared with patients who expressed low levels of this miRNA. We further showed via multivariate analysis that upregulation of this miRNA, along with the clinical stage of the disease, can be used as predictors of OS. Furthermore, although no statistical differences were observed in OS when comparing the expression levels of miR-26a-5p and miR-335-5p, a clear clinical tendency was shown. Thus, analyses of wider cohorts of patients remain necessary to strengthen the clinical utility of both of these miRNAs in the prognosis of CRC.

The inclusion of molecular technology and clinical biomarkers to assess diagnosis and prognosis in CRC holds promise for personalized medicine and the development of innovative management approaches [44]. For this reason, the main perspective of this study is to continue validating the correlation of the expression levels of miR-3065-5p and miR-26a-5p with OS in liquid biopsy using a wide cohort of CRC patients. Moreover, a better understanding of the clinical utility of miR-3065-5p in this study led us to develop PROMIR-C, a predictor score that incorporates the expression levels of this miRNA, and the clinical features associated with the diagnosis and prognosis of CRC. This tool aids clinicians in treatment decisions, ultimately improving patient survival rates. The PROMIR-C score aims to enhance clinical decision-making by providing a comprehensive assessment of CRC prognosis. By integrating both molecular and clinical data, the algorithm offers a personalized risk stratification tool, guiding clinicians in tailoring treatment strategies and optimizing patient outcomes. Additionally, it could facilitate early intervention and monitoring, potentially leading to improved overall survival rates and quality of life for CRC patients.

## 5. Conclusions

The present translational study integrates clinical and molecular features to improve the diagnosis and prognosis of colorectal cancer. The present study validates the usefulness of miR-3065-5p and miR-26a-5p expression levels as biomarkers for CRC diagnosis and prognosis. In addition, we established PROMIR-C, a website tool that integrates miR-3065-5p expression levels as a molecular biomarker related to clinical features and overall survival. This tool was designed and developed to allow clinicians to improve prognostic assessment, and could impact therapeutic decision-making, resulting in better outcomes for CRC patients.

## Figures and Tables

**Figure 1 cancers-16-03649-f001:**
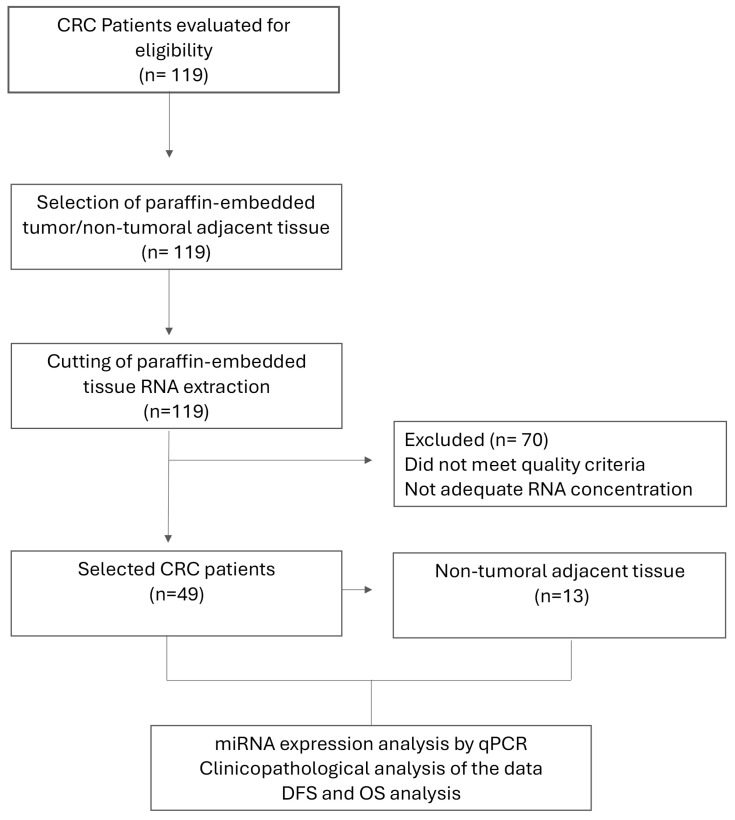
Consort diagram. This diagram describes the population included in this study. Non-tumoral adjacent tissue (*n* = 13), tumoral tissue of CRC patients (TT-CRC) (*n* = 49). CRC: colorectal cancer. DFS: disease-free survival. OS: overall survival. qPCR: quantitative PCR.

**Figure 2 cancers-16-03649-f002:**
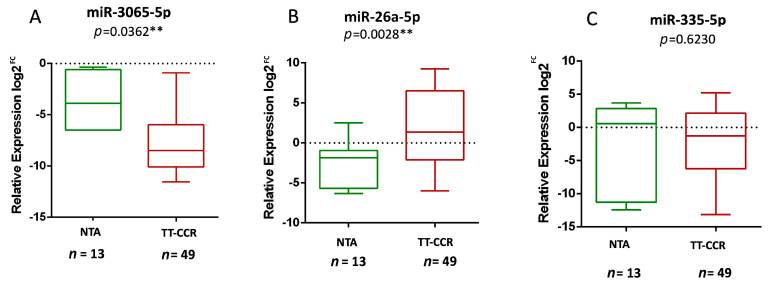
Comparison of miRNA expression levels between non-tumoral adjacent tissue (NTA), and CRC tissues (TT-CRC). The relative expression of (**A**) miR-3065-5p, (**B**) miR-26a-5p, and (**C**) miR-335-5p are shown. The relative expression of miRNAs was calculated using 2^−ΔCT^ analysis using RNU6 levels as an endogenous expression control. ** Statistical significance.

**Figure 3 cancers-16-03649-f003:**
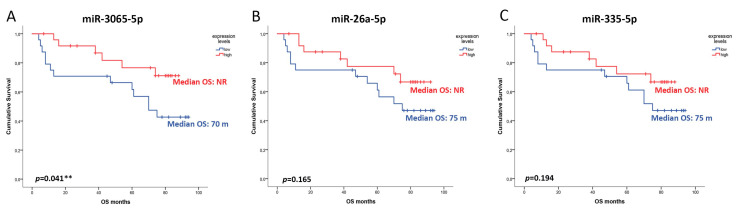
Overall survival (OS) of patients diagnosed with CRC is associated with miRNA levels. Kaplan–Meier estimates overall survival among patients diagnosed with CRC according to miRNA expression levels. (**A**) miR-3065-5p (median expression log 2^−ΔCT^ = 2.5), (**B**) miR-26a-5p (median expression log 2^−ΔCT^ = 1.3), and (**C**) miR-335-5p (median expression log 2^−ΔCT^ = −1.27). Median OS not reached (NR). ** Statistical significance.

**Figure 4 cancers-16-03649-f004:**
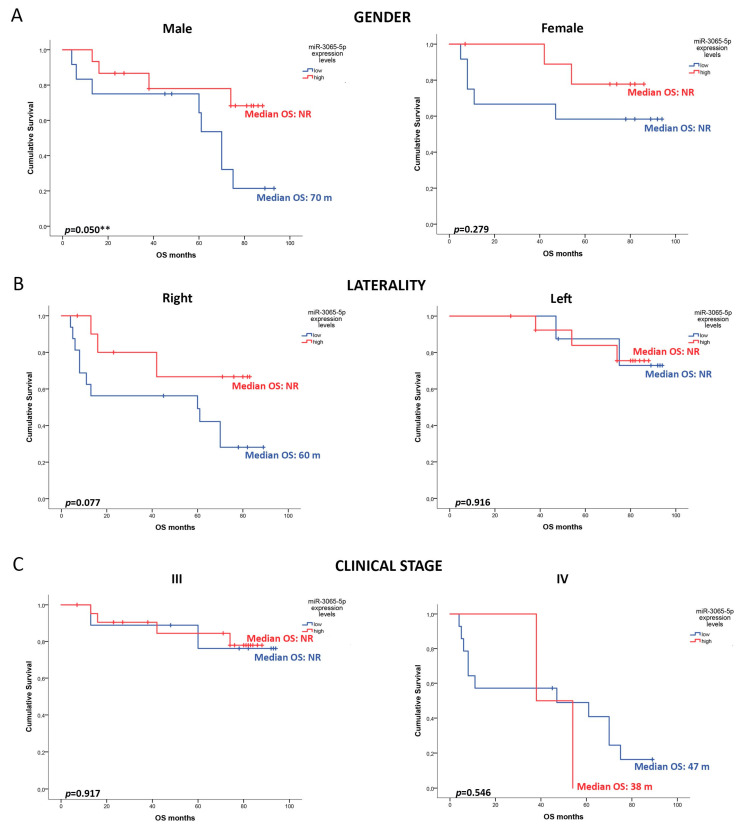
Clinical features and overall survival (OS) association with miR-3065-5p levels. Kaplan–Meier estimates the overall survival of patients with CRC, correlating clinical features and miR-3065-5p expression levels (median expression log 2^−ΔCT^ = 2.5). (**A**) Gender, (**B**) laterality, and (**C**) clinical stage. Median OS not reached (NR). ** Statistical significance.

**Table 1 cancers-16-03649-t001:** Clinicopathological features of the population included.

Clinicopathological Features of CRC Patients	
	*n* = 49 (100%)
Gender	
Male	27 (55%)
Female	22 (45%)
Age (median)	54.5 (19–81)
Clinical stage	
III	33 (65%)
IV	16 (35%)
Tumor location	
Right side	21 (39%)
Left side	28 (61%)
Histological grade	
Well-differentiated	4 (8%)
Moderately differentiated	19 (39%)
Poor differentiated	26 (53%)
Histology	
Mucinous	8 (16%)
Intestinal	41 (84%)
Surgery	
Present	46 (94%)
Absent	3 (6%)
Chemotherapy	
Present	47 (95%)
Absent	2 (5%)

Clinicopathological features of CRC Mexican cohort.

## Data Availability

All data are available on request to Berenice Carbajal López (eb.carbajalopez@gmail.com), Carlos Pérez Plasencia (carlos.pplas@gmail.com), and Eloy Pérez Yepez (eperezy2306@gmail.com).

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
