# Peer review of "miR-3065-5p and miR-26a-5p as Clinical Biomarkers in Colorectal Cancer: A Translational Study"

_cancers, 2024, doi:10.3390/cancers16213649_

Round 1
Reviewer 1 Report
Comments and Suggestions for Authors
In this manuscript the authors evaluate 3 miRNAs as a potential clinical biomarkers in colorectal cancer. This study is interesting and could provide some new information that could be helpful for CRC patients. There are some concerns and propositions I have about this study that are numbered below.
1. It is very difficult to find an endogenous control appropriate for miRNAs because of the way they are synthesized and also their size, especially in cancer. How did the authors determine to use RNU6B as an endogenous control?
2. Figure 1 has a bad quality. Could the authors improve the quality of this figure.
3. The authors mention their previous study where they determine in silica different miRNAs with differential expression in cancer. Could the authors explain in the manuscript why did they chose to work with these 3 miRNAs in this study? Why not with all the miRNAs they mentioned in their previous work?
4. Could the authors explain in more detail the potential use for the site they propose in the manuscript. In this moment it only has miR3065-5p data and there are no explanations for the features. The result shows The score of survival percentage but it´s not clearly stated on the site based on what. The inquiry data? Ando also the site does not explain how is The miRNA adjusted score of survival calculated. Could the authors clarify this. On the other hand, is this site proposed for calculating only this miRNA or is it a work in progress?
Author Response
In this manuscript the authors evaluate 3 miRNAs as a potential clinical biomarkers in colorectal cancer. This study is interesting and could provide some new information that could be helpful for CRC patients. There are some concerns and propositions I have about this study that are numbered below.
Dear reviewer, thank you so much for the feedback on our work. Below you can find the reply to all your comments.
- It is very difficult to find an endogenous control appropriate for miRNAs because of the way they are synthesized and also their size, especially in cancer. How did the authors determine to use RNU6B as an endogenous control?
Answer: Dear reviewer, we appreciate your kind observation. Following your comment, we mistakenly spelled RNU6B as a control instead of RNU6 (U6). We corrected the name of the endogenous control in the manuscript (lines 129 and 255). In addition, we understand your concern about the difficulty of finding the best endogenous control for miRNAs. However, applied biosystems suggest RNU6 as an acceptable endogenous control (more than 290 citations on its website). In the case of colorectal cancer, U6 gene expression has been demonstrated as the most stable control for colorectal adenocarcinoma and even healthy patients (PMID: 24349425). Furthermore, published works from our group on miRNA expression and colorectal cancer have validated U6 as a stably expressed gene in both, healthy and tumoral tissues (PMID: 30983885, 28443472).
- Figure 1 has a bad quality. Could the authors improve the quality of this figure.
Answer: Dear reviewer, we apologize for the mistake, we have changed Figure 1 in the manuscript.
- The authors mention their previous study where they determine in silica different miRNAs with differential expression in cancer. Could the authors explain in the manuscript why did they chose to work with these 3 miRNAs in this study? Why not with all the miRNAs they mentioned in their previous work?
Answer: Dear reviewer, we explained in the introduction (lines 86-89) and discussion (lines 285-289) sections why we selected only these 3 miRNAs.
- Could the authors explain in more detail the potential use for the site they propose in the manuscript. In this moment it only has miR3065-5p data and there are no explanations for the features. The result shows The score of survival percentage but it´s not clearly stated on the site based on what. The inquiry data? Ando also the site does not explain how is The miRNA adjusted score of survival calculated. Could the authors clarify this. On the other hand, is this site proposed for calculating only this miRNA or is it a work in progress?Answer: Dear reviewer, thank you so much for your comments. The clinical features included in PROMIR-C were obtained from a retrospective database from the gastroenterology department of the National Cancer Institute of Mexico. This database contains clinical information (sex, age, tumor location, histological grade, and clinical stage) on 3500 CRC patients. The percentage value for the survival score was calculated based on the association of each clinical feature with the patient’s overall survival (OS). The scores calculated for clinical features associated with poor OS were lower than those associated with a better outcome. The adjusted survival score was determined using miR-3065-5p expression levels (from the 49 patients included in this study) and its association with the OS. At the time, only the miR-3065-5p expression levels were used. However, an additional set of 6 miRNAs is under study to be added to PROMIR-C. Finally, the clinical utility and potential of PROMIR-C are discussed in lines 337-347 of the manuscript.
Reviewer 2 Report
Comments and Suggestions for Authors
By translational study on CRC patients, authors found miR-3065-5p and miR-26a levels discern between CRC and non-tumoral tissues. The study of miRNA biomarkers for CRC prognosis is meaningful, and the webtool PROMIR-C provides overall survival information for CRC patients.
Comments are below.
1. Introduction
Suggest authors separate the whole paragraph into three paragraphs which can be clear for readers to understand.
2. Wrong references NO. 4-NO.14, and NO.16.
Strongly recommend authors review all the references and check their relevance to the context.
From Line 71 to Line 84, miRNA basics and functions in cancers were described. However, none of the references except No. 15 were related to miRNA study!
3. Inappropriate spelling
Line 82: Inappropriate spelling of transcription factors: NF-κB and STAT3.
Line 89: Inappropriate spelling of miR-26a.
Line 123: Inappropriate spelling of 2-∆∆Ct.
Line 223, 226, 231: Inappropriate spelling of miR-3065-5p.
Figure 1: Inappropriate spelling of CRC.
Line 277: Inappropriate spelling of miR-26a.
4. Table 1:
From your PROMIR-C webpage, it found that young patients have lower survival rates than groups older than 31. Maybe authors could list populations of different age groups instead of median age.
5. Results 3.3.
Expression level of three mature miRNAs were analyzed between CRC and non-tumor tissues. Which arm of miR-26a did authors analyze? 3p or 5p?
6. Results 3.5: PROMIR-C.
a. The algorithm, Prognosis miRNAs assessment in Cancer (PROMIR-C), should have more prehensive datasets related to other cancer types and miRNAs. If authors only include 49 CRC patients and one miRNA in the algorithm, the name is inappropriate. Suggest authors have another name related to CRC.
b. The PROMIR-C webpage did not show any information regarding cancer types, patient cohort behind the algorithm.
c. Inappropriate spelling of miR-3065-5p on the webpage.
7. Conclusion
Line 332: PROMIR-C only integrates miR-3065-5p into prediction.
Author Response
By translational study on CRC patients, authors found miR-3065-5p and miR-26a levels discern between CRC and non-tumoral tissues. The study of miRNA biomarkers for CRC prognosis is meaningful, and the webtool PROMIR-C provides overall survival information for CRC patients.
Comments are below.
- Introduction
Suggest authors separate the whole paragraph into three paragraphs which can be clear for readers to understand.
Answer: Dear Reviewer, thank you so much for the recommendation. We divided the introduction section into three paragraphs as you kindly suggested.
- Wrong references NO. 4-NO.14, and NO.16.
Strongly recommend authors review all the references and check their relevance to the context. From Line 71 to Line 84, miRNA basics and functions in cancers were described. However, none of the references except No. 15 were related to miRNA study!
Answer: Dear Reviewer, we apologize for the error in the references that you mentioned. All references included in the manuscript have been checked and corrected.
- Inappropriate spelling
Line 82: Inappropriate spelling of transcription factors: NF-κB and STAT3.
Line 89: Inappropriate spelling of miR-26a.
Line 123: Inappropriate spelling of 2-∆∆Ct.
Line 223, 226, 231: Inappropriate spelling of miR-3065-5p.
Figure 1: Inappropriate spelling of CRC.
Line 277: Inappropriate spelling of miR-26a.
Answer: Dear Reviewer, we apologize for the error in the spelling. We addressed all the inappropriate spelling and corrected as you kindly suggested.
- Table 1:
From your PROMIR-C webpage, it found that young patients have lower survival rates than groups older than 31. Maybe authors could list populations of different age groups instead of median age.
Answer: Dear Reviewer, thank you so much for your observation. At the moment, we only recruited one patient younger than 31 years. Nevertheless, the PROMIR-C age score was calculated with a database that includes more than 100 patients in this group (from a total of 3500). For these reasons, we decided to show in Table 1 only the median age of the 49 patients in the study.
- Results 3.3.
Expression level of three mature miRNAs were analyzed between CRC and non-tumor tissues. Which arm of miR-26a did authors analyze? 3p or 5p?
Answer: Dear Reviewer, we study miR-26a-5p. The spelling of this miRNA was changed in the manuscript and figures.
6. Results 3.5: PROMIR-C.
The algorithm, Prognosis miRNAs assessment in Cancer (PROMIR-C), should have more prehensive datasets related to other cancer types and miRNAs. If authors only include 49 CRC patients and one miRNA in the algorithm, the name is inappropriate. Suggest authors have another name related to CRC.
Answer: Dear reviewer, thank you for your suggestion. We agree that, for this work, we only validated miRNA expression levels to CRC. However, our group is currently developing studies that aim to expand the number of miRNAs to be included in PROMIR-C. In addition, one of the perspectives of this study is to validate the web tool in other types of cancer. Therefore, we decided to use the name of PROMIR-C.
The PROMIR-C webpage did not show any information regarding cancer types, patient cohort behind the algorithm.
Answer: Dear reviewer, we added information into the PROMIR-C webpage about the patient´s cohort in the survival score calculation.
Inappropriate spelling of miR-3065-5p on the webpage.
Answer: Dear reviewer, we corrected the spelling on the webpage.
- Conclusion
Line 332: PROMIR-C only integrates miR-3065-5p into prediction.
Answer: Dear reviewer, thank you so much for your observation. We modified the conclusion section (lines 352-356).
Reviewer 3 Report
Comments and Suggestions for Authors
The main question addressed by the research is by evaluating the clinical utility of miRNAs’ expression levels in colorectal cancer (CRC) from 49 CRC patients, to identify the potential miRNA which can be used as biomarkers for CRC diagnosis and prognosis.
The data was of high quality and clearly illustrated. The conclusions are consistent with the evidence and arguments presented. The references are appropriate. Overall, this work is suitable for publication after addressing a few minor concerns.
comments:
1. If the patient samples include clinical stage I-II, the potential miRNA biomarker for early CRC detection maybe can be found, that would be more useful by combining the PROMIR-C system. Do you have clinical stage I-II samples and have you considered the evaluation?
2. But in the PROMIR-C website the clinical stage including 0-II, how can this have predicted without data in the paper?
3. Consisting of the patient’s number with number or word (like sixty-two patients, 119 patients).
Author Response
The main question addressed by the research is by evaluating the clinical utility of miRNAs’ expression levels in colorectal cancer (CRC) from 49 CRC patients, to identify the potential miRNA which can be used as biomarkers for CRC diagnosis and prognosis.
The data was of high quality and clearly illustrated. The conclusions are consistent with the evidence and arguments presented. The references are appropriate. Overall, this work is suitable for publication after addressing a few minor concerns.
comments:
- If the patient samples include clinical stage I-II, the potential miRNA biomarker for early CRC detection maybe can be found, that would be more useful by combining the PROMIR-C system. Do you have clinical stage I-II samples and have you considered the evaluation?
Answer: Dear reviewer, thank you so much for your kind comments. Patients in clinical stage I-II were not included in this study. However, their inclusion remains as a perspective for PROMIR-C validation to survival score calculation in the early stages of CRC.
- But in the PROMIR-C website the clinical stage including 0-II, how can this have predicted without data in the paper?
Answer: Dear reviewer, the clinical features included in PROMIR-C were obtained from a retrospective database from the gastroenterology department of the National Cancer Institute of Mexico. This database contains clinical information (sex, age, tumor location, histological grade, and clinical stage 0-IV) of 3500 CRC patients. The percentage value for the survival score was calculated based on the association of each clinical feature with the patient’s overall survival (OS). The scores calculated for clinical features associated with poor OS were lower than those associated with a better outcome. The adjusted survival score was determined using miR-3065-5p expression levels (from the 49 patients included in this study, stage III-IV) and its association with the OS.
- Consisting of the patient’s number with number or word (like sixty-two patients, 119 patients).
Answer: Dear reviewer, we apologize for the inconsistency. We change the numbers by words (lines 32, 108, 148, 156 and 324).
Reviewer 4 Report
Comments and Suggestions for Authors
Authors evaluated the clinical utility of miRNAs’ expression levels in CRC, such as miR-26a and miR-3065-5p. The study has clinical value, but the authors need more careful preparation.
1. The control group sample setting needs more detailed description.
a. The adjacent tissue of the same patient’s tumor tissue is the most used and biologically significant control group.
b. Normal tissues, please provide basic information of the group, such as age.
2. Calculation formula and method of miRNA expression level
a. 2-ΔCT is used in the method, which becomes 2ΔΔCt in the annotation of Figure 2.
b. Ct value for quantitative detection of miRNAs.
c. Please indicate the miRNA expression levels in Figure 3 and Figure 4.
3. In the discussion section, the authors failed to compare and analyze the results well with published research papers.
a. In paper ‘Tumor-suppressive miR-26a and miR-26b inhibit cell aggressiveness by regulating FUT4 in colorectal cancer’, miR-26a expression was significantly decreased in 38 pairs of CRC tissues compared with the corresponding adjacent noncancerous tissues by qRT-PCR analysis.
Author Response
Authors evaluated the clinical utility of miRNAs’ expression levels in CRC, such as miR-26a and miR-3065-5p. The study has clinical value, but the authors need more careful preparation.
- The control group sample setting needs more detailed description.
- The adjacent tissue of the same patient’s tumor tissue is the most used and biologically significant control group.
- Normal tissues, please provide basic information of the group, such as age.
Answer a and b: Dear reviewer, please accept our apology for the misunderstanding. As you commented, the adjacent non-tumoral tissue of the same patient is a better control group. For this reason, we used this kind of tissue as the reference group in our analysis. We clarified this on lines 112-113 at the 2.2 material and methods section.
For this study, we did not use normal tissues.
- Calculation formula and method of miRNA expression level
- 2-ΔCT is used in the method, which becomes 2ΔΔCt in the annotation of Figure 2.
Answer: Dear reviewer, thank you so much by the observation, we corrected the annotation of Figure 2 (line 254).
b. Ct value for quantitative detection of miRNAs.
Dear reviewer, we present a table that contains the log2 of 2-ΔCT and CT values (miRNA/Constitutive gene) median value:
|
||||||||||||||||||||
|
c. Please indicate the miRNA expression levels in Figure 3 and Figure 4.
Answer: Dear reviewer, we address your request in figure 3 legend (lines 266-267) and figure 4 legend (line 272).
- In the discussion section, the authors failed to compare and analyze the results well with published research papers.
a. In paper ‘Tumor-suppressive miR-26a and miR-26b inhibit cell aggressiveness by regulating FUT4 in colorectal cancer’, miR-26a expression was significantly decreased in 38 pairs of CRC tissues compared with the corresponding adjacent noncancerous tissues by qRT-PCR analysis.
Answer: Dear reviewer, we recognize that miRNAs research may show contradictory data. However, our argument that miR-26a-5p is an onco-miR is based on a large number of previously published evidence (DOI: 10.1038/s41598-024-56361-2; DOI: 10.2217/bmm-2022-0861; DOI: 10.3892/ol.2022.13207; DOI: 10.3390/cancers14020462; DOI: 10.1016/j.canlet.2021.08.017; DOI: 10.1186/s12935-019-0802-5), which coincides with our observations that this miRNA increases the expression levels in colon cancer-derived tissue compared to adjacent healthy or non-tumoral tissue. These reports have been made with different groups of study worldwide, including Mexican population.
Round 2
Reviewer 1 Report
Comments and Suggestions for Authors
I thank the authors for addressing all my concerns and making changes to the manuscript. In my opinion the program they developed could have benefited from a little more developing before publishing it. I would have liked to have seen this program fully equipped with miRNAs and functioning. Nevertheless, I consider that the manuscript is now sufficiently improved and ready to be published.
Author Response
Dear Reviewer, Thank you so much for your kind comments. We understand your opinion about PROMIR-C; in fact, a perspective of this study is to strengthen PROMIR-C by adding other miRNAs.
Reviewer 2 Report
Comments and Suggestions for Authors
Authors responded to comments and most of them were accepted by the reviewer. Minor modifications might need to be made as follows.
1. Figure 2 and 3.
From the legend, the relative expression of miRNAs(FC) was calculated using 2-∆CT. In the y axis, it was shown as log2. Please specify fold change(FC) calculation as FC=2-∆CT or the numbers in y axis were shown as log2FC.
Same as in legend of figure 3.
2. PROMIR-C algorithm and webpage
a. How many patients were included when generating the algorithm? From the webpage annotation #1, it is shown as 3,500. However, no clear descriptions were mentioned in neither Methods nor Results. From the manuscript, all I know is 49 patients being studied out of 119 candidates. Please clarify patient numbers in the webpage.
b. Cancer type. Please clarify the cancer type in the webpage. The title of “PROMIR-C”, “Clinical Features”, “miRNAs Adjustment”, and “Overall Survival Score” have nothing to do with colorectal cancer.
I understand that authors have ongoing studies/plans for other cancer types and miRNAs. Since now the webpage can only provide CRC patient prognosis, please show current cancer type in a clear way instead of mentioned in the annotation.
Author Response
Dear Reviewer, thank you so much for your accurate comments. Here you can find our reply to your observations.
- Figure 2 and 3.
From the legend, the relative expression of miRNAs(FC) was calculated using 2-∆CT. In the y axis, it was shown as log2. Please specify fold change(FC) calculation as FC=2-∆CT or the numbers in y axis were shown as log2FC.
Same as in legend of figure 3.
Answer: Dear reviewer, thank you so much for your observations. We already made the changes in Figures 2 and 3.
- PROMIR-C algorithm and webpage
- How many patients were included when generating the algorithm? From the webpage annotation #1, it is shown as 3,500. However, no clear descriptions were mentioned in neither Methods nor Results. From the manuscript, all I know is 49 patients being studied out of 119 candidates. Please clarify patient numbers in the webpage.
Answer: Dear reviewer, we modified the section “2.6 Algorithm for prognosis” of material and methods, including the number of patients for clinical features and molecular adjustment (lines 146-154).
- Cancer type. Please clarify the cancer type in the webpage. The title of “PROMIR-C”, “Clinical Features”, “miRNAs Adjustment”, and “Overall Survival Score” have nothing to do with colorectal cancer.
I understand that authors have ongoing studies/plans for other cancer types and miRNAs. Since now the webpage can only provide CRC patient prognosis, please show current cancer type in a clear way instead of mentioned in the annotation.
Answer: Dear reviewer, we have changed the PROMIR-C web page adding a section that clearly specifies the type of cancer that is currently analyzing (colorectal cancer).
Reviewer 4 Report
Comments and Suggestions for Authors
No further suggestions.
Author Response
Dear Reviewer, thank you so much for taking the time to review our manuscript.